



# The iFlow Modelling Framework v2.4. A modular idealised process-based model for flow and transport in estuaries.

Yoeri M. Dijkstra[1], Ronald L. Brouwer[1,2], Henk M. Schuttelaars[1], and George P. Schramkowski[1,2]

[1]Delft Institute of Applied Mathematics, Delft University of Technology, P.O. Box 5031, 2628 CD Delft, The Netherlands
[2]Flanders Hydraulics Research, Berchemlei 115, 2140 Antwerp, Belgium

*Correspondence to:* Y.M. Dijkstra (Y.M.Dijkstra@tudelft.nl)

**Abstract.** The iFlow modelling framework allows for a systematic analysis of the water motion and sediment transport processes in estuaries and tidal rivers and the sensitivity of these processes to model parameters. iFlow has a modular structure, making the model easily extendible. This allows one to use iFlow to construct anything from very simple to rather complex models.

The iFlow core is designed to make it easy to include, exclude or change model components, called modules. The core automatically ensures modules are called in the correct order, inserting iteration loops over groups of modules that are mutually dependent. The iFlow core also ensures a smooth coupling of modules using analytical and numerical solution methods or modules that use different computational grids.

iFlow includes a range of modules for computing the hydrodynamics and suspended sediment dynamics in estuaries
and tidal rivers. These modules employ perturbation methods, which allow for distinguishing the effect of individual forcing terms in the equations of motion and transport. Also included are several modules for computing turbulence and salinity. These modules are supported by auxiliary modules, including a module that facilitates sensitivity studies.

Additional to an explanation of the model functionality, we present two case studies, demonstrating how iFlow facilitates the analysis of model results, the understanding of the underlying physics and the testing of parameter sensitivity. A
comparison of the model results to measurements show a good qualitative agreement.

## 1   Introduction

The dynamics of estuaries and tidal rivers is characterised by the complex interplay of mutually interacting processes related to the water motion (i.e. tidal propagation, river run-off), salinity and sediment dynamics, transport of nutrients and bathymetric changes. In many estuaries and tidal rivers these processes are subject to constant change due to human
interventions, such as dredging and canalisation, or by natural changes, such as sea level rise or changing river discharge. These changes may lead to practical problems. Focussing on the hydrodynamics and sediment dynamics, examples are increasing risks of flooding related to tidal amplification or reflection (e.g. Friedrichs and Aubrey, 1994; Winterwerp et al., 2013; Schuttelaars et al., 2013) and deteriorating ecosystems due to a decreased light penetration caused by increasing suspended sediment concentrations (e.g. Colijn, 1982; Cloern, 1996; De Jonge et al., 2014). Many systems face several





simultaneous natural and anthropogenic changes, which each affect multiple processes. Therefore the understanding of these processes and their interrelations through models, in combination with observational evidence, is of paramount importance in anticipating the effect of future natural and anthropogenic change.

A wide range of process-based models has contributed to the present-day understanding of flow and transport pro-
cesses. These models range from linear one-dimensional along channel models to non-linear three-dimensional numer-
ical models. One way of classifying models is to describe their position in the spectrum ranging from *exploratory* to
*complex* models (Murray, 2003). On one end of this spectrum, exploratory, or idealised, models typically include a limited
number of processes that are thought to be important for the particular phenomenon that is studied. These models come
in many forms, ranging from one-dimensional to three-dimensional and from analytic to numeric. The common property
of these models is their excellent ability to quickly investigate the sensitivity to parameter variations and to systematically
study individual physical processes. Since they are often purpose-build, the applied solution techniques do not allow for
an easy extension to more processes or complex model domains. Therefore the comparison between these models and
real-life systems has to be qualitative and one carefully needs to consider the effect of the underlying assumptions. On
the other side of the spectrum, complex models aim at a quantitative comparison of the model results with observations
in a wide range of real systems. This requires the implementation of most known processes and their mutual interactions
through state-of-the-art parametrisations. As a result, such models are typically numerical and non-linear, and compu-
tation times are relatively long. This makes complex models less suitable for identifying the essential processes and
conducting extensive sensitivity studies.

The aim of the iFlow modelling framework presented in this paper is to combine the strengths of both approaches
identified above. That is, to represent the complex processes and interactions contained in complex models, while retain-
ing the ability to analyse these processes and study their sensitivity. iFlow is a width-averaged model for hydrodynamics
and sediment transport processes in single-branch estuaries and tidal rivers, focussing on global estuarine processes.
Within this context, the model is able to cover a wide range of complexity, reaching out to both the idealised and com-
plex model types. This requires a structured and systematic approach. This approach starts by taking an established
and well-understood exploratory model that solves for a specific subset of hydro- and sediment dynamical processes
using a combination of analytical and semi-analytical solution methods. The power of iFlow lies in its ability to extend
this basic model by increasingly more complex and realistic interactions, which can either be included or excluded in the
model depending on the application. These extensions can often only be resolved numerically and sometimes require
iterative methods. The model thus naturally consists of a set of coupled and mutually interacting components that solve
for different processes using different solution techniques. These model components are called *modules* in iFlow. Mod-
ules form code-independent entities that can be developed independently and can be easily added to the model without
requiring changes to other modules. The iFlow core takes care of the coupling of modules through a simple standardised
input/output protocol, thus facilitating interactions between processes in different modules. This allows for a natural de-
velopment of the model by implementing new processes or different implementations of already existing ones, motivated
by the needs for the application at hand.





iFlow currently includes several modules that allow for the computation of the flow and suspended sediment transport. Most of these modules focus on identifying the effect of individual processes and to this end use a perturbation approach. This approach has been successfully applied before in the context of estuarine research by e.g. Ianniello (1977, 1979); Chernetsky et al. (2010); Cheng et al. (2010); Wei et al. (2016). The perturbation approach is used to identify processes that balance at different orders of magnitude. Under suitable assumptions of weakly non-linear flow, the leading-order flow and sediment balances reduce to linear equations describing the propagation of the tide and tidal re-suspension of sediment. These balances match classical exploratory model results (e.g. Prandle (1982); Hansen and Rattray (1965); Friedrichs and Aubrey (1994) and references therein). Non-linear processes and other processes that are not of leading order are however not neglected. Rather, linear estimates of the non-linear processes are taken into account at the first and higher orders. Because of the linearity, the effects of each process on the flow and sediment concentration can be evaluated separately. In this way, the fully non-linear solution can theoretically be approximated to any degree of accuracy, while the effects of individual processes and interactions can still be analysed. Practically, it turns out that the qualitative properties of the solution are often described by only a limited set of orders and processes.

Summarising, the iFlow philosophy revolves around three central ideas:

1. The model is easily extendible by new processes.
2. The model allows for the combination of different solution methods for different processes, including analytical and numerical solution methods.
3. It is possible to identify the effects of individual physical forcing mechanisms and interactions.

We elaborate on the way these ideas are implemented in Section 2, which discusses the modular model structure in detail using a basic example involving four modules. A list of all modules currently included in the model in provided at the end of this section. Subsequent sections focus on the specific modules that form iFlow's current functionality. Section 3 presents the model domains and numerical grids currently allowed. Section 4 then provides a discussion of the modules for hydrodynamics and sediment dynamics, focussing on the assumptions and options in these modules. A short outline of the other main modules, including the various turbulence closures, salinity models and sensitivity module is provided in Section 5. Two examples of model applications, to the Yangtze and Scheldt Rivers, are presented in Section 6. This paper finalises with conclusions and a guide to the code availability in Sections 7 and 8. While this paper provides an overview of the model features and methods, an in-depth user-manual and a full technical description of the model is provided in the supplementary material.

## 2 Modular structure

In order to satisfy the three criteria set in the introduction (extendibility, interchangeability and ease of analysis) the structure of iFlow has to be modular. Modules are separate model entities that implement certain physical processes or perform auxiliary tasks, such as plotting or initiating a sensitivity study. A module may use any approach to obtain the





required variables, for example by solving a set of equations directly, by loading measured or modelled data from a file or even by linking to another modelling suite. Modules are code-independent, meaning that the interaction between different modules is only on input and output level, not on code level. This allows an independent development of modules by different developers, while ensuring seamless interaction between different modules. It also allows easy interchangeability

of modules that compute the same variables, but that differ in the physical processes taken into account or the type of implementation used.

Depending on the problem at hand, users can select which variables to save, what physical processes to include and what auxiliary tasks to perform by selecting a set of modules. These modules are listed in an *input file*, together with the input parameters required by these modules. Upon the start of a simulation, iFlow will read the input file and start an

automated two-step process: ordering the modules into a *call stack* and then calling the modules in this order. Below, these steps are explained and illustrated using the example displayed in Figure 1, which gives a simplified demonstration of the computation of the leading-order flow velocity through a set of four interacting modules.

## 2.1  Building the call stack

As a first step, iFlow reads the input file (Figure 1a) and compiles a list of the modules. In order to determine the order

in which to call these modules, iFlow needs information on the input required and output returned by each module. This information is documented in a *registry file* (Figure 1b), which is provided with the modules and does not need to be given on input. The call stack is made by matching the output provided by each module to the input required by the other modules such that the required input is available at the moment a module is called. Here we will discuss the way in which iFlow uses the input and registry files to construct a call stack. More detailed information on the meaning and format of

the variables listed in these files can be found in the manuals, which are attached as supplementary material.

The input file lists four modules with a specific task each: *RegularGrid* for making a grid, *Geometry2DV* for setting the geometry, *Hydrolead* for computing the leading-order hydrodynamics and *KEFitted* as turbulence closure. At the end, the input file lists the variables that are required by the user, e.g. for saving or plotting, here these variables are the leading-order velocity $u^0$ and eddy viscosity $A_\nu$ (more information on these variables and the underlying equations will

be provided in Sections 4 and 5). The registry file (Figure 1b) contains the same modules with their input and output variables. Using the registry file, iFlow assesses that the output of the module *Hydrolead*, $u^0$, and of *KEFitted*, $A_\nu$ are needed to obtain the required variables. iFlow then constructs the call stack, by determining the modules needed before calling *Hydrolead* and *KEFitted*. Focussing on *Hydrolead*, it follows from the registry that this module requires nine input variables. These variables may be provided in the input file, by the output of other modules or in a configuration file (not

shown here, see the manual for details). Three of these input variables, $A^0$, phase$^0$ and $Q^0$ are provided in the input file, while the other six follow from the output of other modules. By matching all the input for and output of the four modules, iFlow constructs the call stack depicted in Figure 1c.

The call stack shows a loop between *Hydrolead* and *KEFitted*, which is necessary as both require each other's output as input. This interdependency is resolved by defining *KEFitted* as an *iterative module*. Behind the keyword *inputInit* in





```
##################
## Input File   ##
##################
## Grid ##
module  numerical2DV.RegularGrid
xgrid   equidistant     100
zgrid   equidistant     50
fgrid   integer         2

## Geometry ##
module  analytical2DV.Geometry2DV
L       150000
B0      type    functions.Polynomial
        C       -1.4e-6 7.5e-3 9.8e2
H0      type    functions.Constant
        C0      10

## Hydrodynamics ##
module  numerical2DV.HydroLead
A0      0 1.5
phase0  0 0
Q0      100

## Turbulence ##
module  analytical2DV.KEFitted
profile uniform
z0*     0.005
Avmin   1.e-6

Requirements    u0 Av
```

(a) Input file

```
##################
## Registry     ##
##################
module          RegularGrid
input           xgrid zgrid fgrid H B L
output          grid

module          Geometry2DV
input           H0 B0 L
output          H B L

module          HydroLead
input           A0 phase0 Q0 grid
                H B L Av roughness
output          u0
submodules      tide river

module          KEFitted
inputInit       profile z0* Avmin grid
input           profile z0* Avmin grid u0
output          Av roughness
iterative       True
```

(b) Registry file

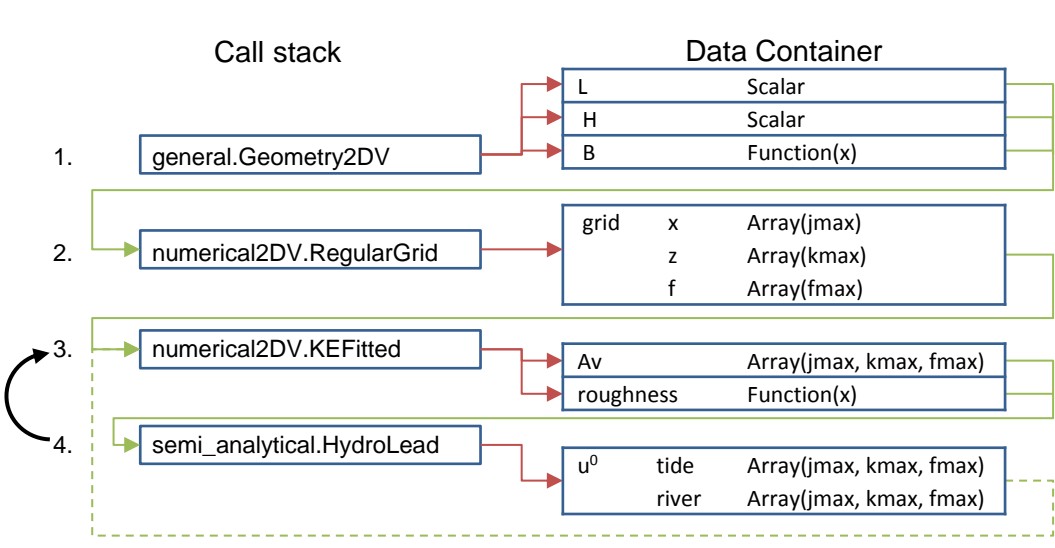

(c) Call stack and communication with the *DataContainer*

**Figure 1.** Basic example of an input (a) and registry (b) files for a model with four modules. The core uses the input and registry files to make a call stack (c) with the correct order of the modules. The modules module output container is stored in the data container to be used as input to other modules.





the registry of *KEFitted* it can be seen that this module does not require the flow velocity $u^0$, computed by *Hydrolead*, for its first run. In subsequent runs of the iteration, $u^0$ is required. iFlow recognises the interdependency and constructs the smallest possible iteration loop, here involving the two interdependent modules only. The number of iterations follows dynamically from a convergence criterion that should be implemented in the iterative module *KEFitted*.

As a consequence of the way that iFlow constructs the call stack, the model will not use modules that are not needed to compute the required variables. A notification of this is given when running a simulation. Similarly, a notification is given if the call stack cannot be completed, because certain input variables are missing.

The example discussed here can easily be extended, e.g. by adding modules for computing additional variables, adding auxiliary modules for saving the output or plotting it, or by adding modules that compute one or more variables that are

now provided in the input file. To allow for more flexibility, the input and output files allow for a number of additional options that are beyond the scope of this paper, including submodules and input-dependent output requirements. Details on this are provided in the iFlow manual in the supplementary material.

### 2.2   Running and data management

After construction of the call stack, the modules are called sequentially in the order defined by the call stack. As modules

are required to be code-independent, they are not allowed to communicate directly with each other. Instead, the iFlow core regulates the distribution of the required input data and collection the resulting output. The management of this data is facilitated by the *DataContainer* in the iFlow core. It collects the module's output upon completion and handles the input data requests by each module, see Figure 1c. To simplify the interchangeability of modules and the analysis of data, the *DataContainer* supports various data types and data decompositions as is discussed more elaborately below.

As different modules used within one simulation can have widely different degrees of complexity and are allowed to use different solution methods, the requested input and resulting output data can be of different types. These data types may include scalars, multi-dimensional arrays and analytical function descriptions. In our example, *Geometry2DV* sets a constant depth $H$, which is saved as a scalar value (see also Figure 1c). Other implementations of the depth allow for depths varying over the horizontal $x$-coordinate according to prescribed analytical functions or data on a grid. This

difference in the way the depth is prescribed should not influence the functioning of other modules. The *DataContainer* allows this by providing a uniform interface to all data types. This means that there is one command for a module to retrieve $H$ (or any other variable) regardless of the underlying data type. The *DataContainer* handles this command based on the data type. For example, the module *RegularGrid* requests $H$ on grid points. If $H$ is stored as a scalar, the *DataContainer* automatically extends this scalar value to all requested points. If $H$ is stored as an analytical function description, this

function is evaluated at the grid points. Data stored on numerical grids may as well be used as input to analytical functions. If the numerical data is requested at other coordinates than the grid points, the *DataContainer* automatically interpolates this data to the requested coordinates. Similarly, a module can access the derivative of a variable. iFlow sees whether an analytical function or numerical data for this derivative is provided and will automatically perform numerical differentiation necessary.





Since iFlow is designed to improve the understanding of physical processes, modules may offer decompositions of data into contributions resulting from different physical components. The method of decomposition is the responsibility of individual modules, an example using the perturbation method will be discussed in Section 4 for the hydrodynamics and sediment dynamics modules. Within iFlow's philosophy, It should be possible to interchange these modules by others that do not make decompositions or make decompositions in different components, without affecting other modules. The *DataContainer* supports this using sub-variables. This is illustrated in Figure 1c for the flow velocity variable $u^0$. This has contributions induced by the tide and by the river discharge, such that the sum of both yields the total flow velocity $u^0$. The turbulence model *KEFitted* does not require this decomposition and does not necessarily need to be aware that such a decomposition exists. It can therefore simply request $u^0$ and iFlow will automatically sum the tide and river contributions. Alternatively a module may request a list of all the sub-variables of $u^0$ and request each of these contributions separately.

The *DataContainer* as interface for different data types and decompositions of data thus ensures that modules with different (e.g. analytical and numerical) solution methods can be used together. Additionally, a module can easily be replaced by a different module that results in the same output variables through other processes, without requiring any changes to other modules.

## 2.3 iFlow standard modules

The iFlow modelling framework includes a number of standard modules that may be used to simulate and analyse the water motion and sediment dynamics in estuaries and tidal rivers. Together, the standard modules provide a full model for hydrodynamics and sediment dynamics that may be used in different combinations to model various levels of complexity. The modules are organised in four packages: `general`, `analytical2DV`, `numerical2DV` and `semi_analytical2DV`, containing auxiliary modules and modules using analytical, numerical or semi-analytical (i.e. largely analytical, with numerical components) solution methods respectively. All included standard modules and the location where they can be found are listed in Table 1.

A short introduction to many of these modules is provided in Sections 3-5. Section 3 introduces the standard module for geometry and grid. The standard modules for hydrodynamics and sediment dynamics are introduced in Section 4. A short explanation of other modules related to salinity, turbulence, reference level and sensitivity analyses is given in Section 5.

## 3 Model domain and grid

The iFlow core has a flexible definition of the model dimensions that allows for anything from one-dimensional to three-dimensional models. In this paper we will discuss the standard modules in iFlow version 2.4, which are only for a two-dimensional width-averaged (2DV) model. The along-channel axis is defined as the $x$-coordinate and the vertical axis is defined as the $z$-coordinate. The length of the estuary is thus measured by following the channel between the seaward boundary $x = 0$ and the landward boundary $x = L$ and can be freely chosen. The width, $B$, and bed level, $H$, of the





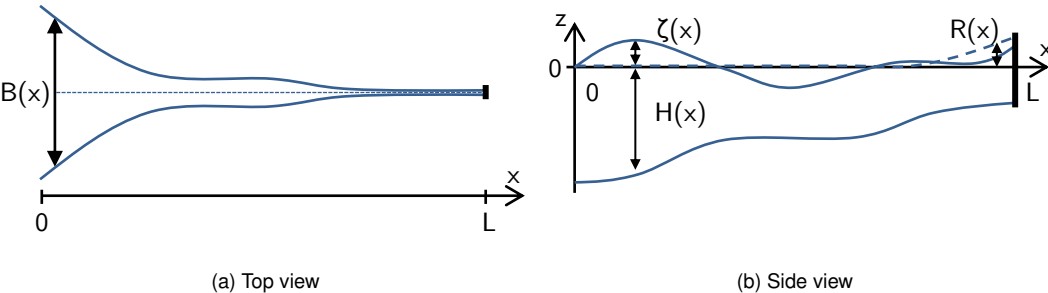

(a) Top view       (b) Side view

**Figure 2.** Model domain. The model is two-dimensional in along-channel ($x$) and vertical ($z$) direction and is width-averaged. The depth and width are allowed to vary smoothly with $x$.

estuary can be provided as arbitrary smooth functions of $x$, see Figure 2. The bed level $H$ is relative to mean sea level at the mouth (MSL) defined at $z = 0$. iFlow contains several built-in functions describing the depth and width, including polynomial and exponential functions. These functions and their derivatives are computed analytically to obtain maximum accuracy. Alternatively, the depth and width may be provided as a list of numerical data on a grid.

The surface level relative to $z = 0$ is denoted by $R + \zeta$, where $R$ denotes the reference level and $\zeta$ denotes the surface elevation. By default, $R = 0$, but the use of a non-zero reference level is required if the river bed is above MSL over parts of the domain. A non-zero reference level is also useful when the mean surface elevation above MSL becomes of the same order of magnitude as the depth. In such cases, the bed level alone $H$ is not a good estimate of the mean water depth. The reference level $R$ is a quick estimate of the local mean surface level, such that $H + R$ is always positive and is

a good approximation of the mean water depth, thereby significantly improving the model accuracy. More details on the computation of $R$ are provided in Section 5.3

    As discussed in Section 2, iFlow modules can use a combination of analytical and numerical solution methods. Each of these modules and solution methods may or may not require a numerical grid and grids may serve different purposes. Apart from using grids for (partly) numerical computations, a grid may be used to save or plot variables as numerical

data. iFlow allows for using different grids in different modules or omitting a grid altogether. As a result, computations in different modules may use grids with different resolutions and the output may be stored on yet a different grid. Automatic linear interpolation of data between different grids ensures a smooth coupling of modules using different grids. Here the standard grid module of iFlow, called RegularGrid is discussed. RegularGrid defines two grids: one computational grid used in all numerical modules and one potentially different output grid. In many cases it is useful to have an output grid

with a low resolution to limit the size of the output data, while using a higher resolution computational grid for the benefits of the model accuracy. iFlow grids are curvi-linear and may be non-equidistant in both the $x$ and $z$-direction. More details can be found in the iFlow manuals, attached as a supplement.





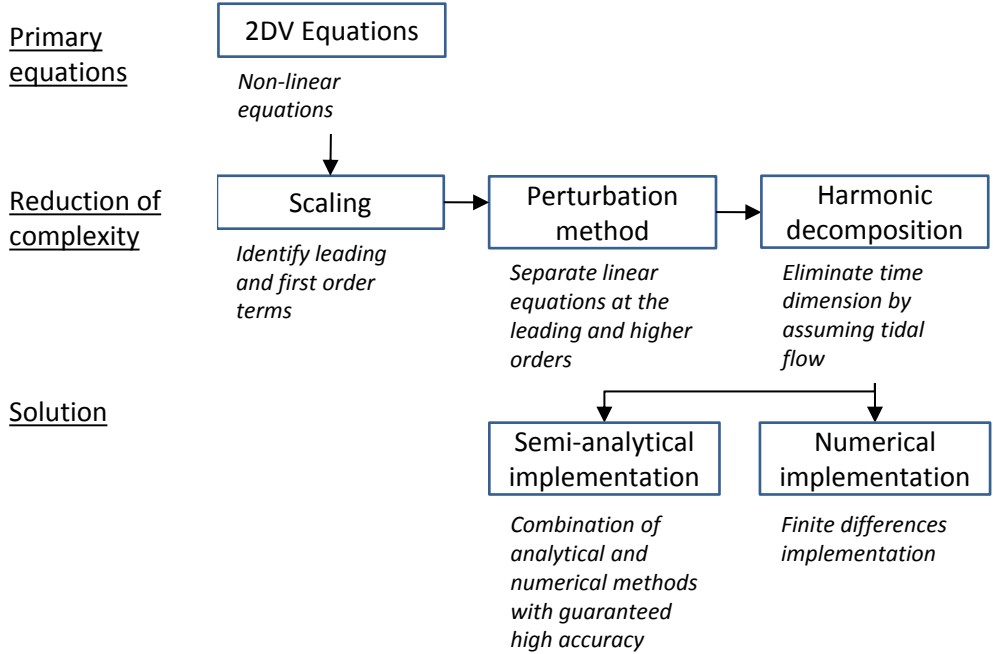

**Figure 3.** Flow diagram outlining the main steps taken in the derivation of the implemented equations for hydrodynamics and sediment dynamics. The fully non-linear width-averaged equations are taken through several steps of analysis to reduce the complexity of the system. Then two implementations of this reduced system are made, each with their own advantages and disadvantages.

## 4   Equations and solution methods for hydrodynamics and sediment dynamics

The standard modules for computing the hydrodynamics and sediment dynamics fit particularly well in the iFlow philosophy as they allow for a separate analysis of the physical contributions to the total result. These analysis properties result from the perturbation approach that is used to solve the continuity, momentum and sediment balances.

The steps taken in the perturbation analysis are listed in Figure 3, which also forms the outline of this section. After presenting the basic width-averaged equations (Section 4.1), these are reduced in complexity via a scaling analysis (Section 4.2), perturbation approach (Section 4.3) and harmonic decomposition (Section 4.4). Finally we will discuss the two solution methods (semi-analytical and fully numerical) implemented in the standard modules (Section 4.5). Throughout the whole section we will focus on the assumptions made in this procedure and the way in which this approach helps to analyse the model result.

### 4.1   Equations

The water motion is described by the Reynolds-averaged width-averaged shallow water equations, that solve for the water level elevation $\zeta(x,t)$, horizontal velocity $u(x,z,t)$ and vertical velocity $w(x,z,t)$. Here $t$ denotes time. We neglect





the effects of Coriolis and assume that density variations are small compared to the average density, allowing for the Boussinesq approximation. The resulting momentum equation reads (e.g. Chernetsky et al., 2010)

$$u_t + uu_x + wu_z = -g\zeta_x - g \int_z^{R+\zeta} \frac{\rho_x}{\rho_0} \, d\tilde{z} + (A_\nu u_z)_z, \tag{1}$$

with a no-stress boundary condition at the free surface and a partial slip condition at the bed

$$A_\nu u_z = 0 \qquad\qquad\qquad \text{at } z = R + \zeta, \tag{2}$$

$$A_\nu u_z = s_f u, \text{ or } u = 0 \qquad\qquad\qquad \text{at } z = -H. \tag{3}$$

For $s_f \to \infty$, the partial slip condition reduces to a no-slip condition. The partial slip becomes a quadratic bottom friction law if $s_f$ is made dependent on the local velocity (see also Section 5.1).

The depth-integrated continuity equation reads

$$\zeta_t + \frac{1}{B} \left( B \int_{-H}^{R+\zeta} u \, dz \right)_x = 0, \tag{4}$$

with boundary conditions

$$\zeta = A \qquad \text{at } x = 0, \tag{5}$$

$$B \int_{-H}^{R+\zeta} u \, dz = Q \qquad \text{at } x = L. \tag{6}$$

Finally the continuity equation reads

$$w_z + \frac{1}{B} (Bu)_x = 0, \tag{7}$$

with a kinematic boundary condition at the free surface and a non-permeability condition at the bed

$$w = \zeta_t + u\zeta_x \qquad \text{at } z = R + \zeta. \tag{8}$$

$$w + uH_x = 0 \qquad \text{at } z = -H. \tag{9}$$

In these equations $g$ is the acceleration of gravity, $\rho$ is the density with reference density $\rho_0$. The vertical eddy viscosity is denoted by $A_\nu$ and $s_f$ is a partial slip roughness parameter. $A = A(t)$ is the time-dependent tidal forcing at the seaward boundary $x = 0$ and $Q$ is the river discharge. The subscripts $x$, $z$ and $t$ in the equations denote derivatives with respect to these dimensions. The background horizontal eddy viscosity $A_h$ has been neglected in (1).

The sediment dynamics is described by the width-averaged sediment mass balance equation, which solves for the sediment concentration $c(x, z, t)$ in the model domain. The sediment is assumed to consist of non-cohesive, fine particles that have a uniform grain size (i.e. constant settling velocity) and are transported primarily as suspended load. At the





surface we do not allow transport of sediment through the water surface and at the bottom we assume that the diffusive flux equals the erosion flux $E$. The resulting equation is (e.g. Chernetsky et al., 2010)

$$c_t + uc_x + wc_z = w_s c_z + \frac{1}{B}(BK_h c_x)_x + (K_v c_z)_z,$$  (10)

with vertical boundary conditions

$w_s c + K_v c_z - K_h c_x \zeta_x = 0$       at $z = R + \zeta,$  (11)

   $-K_v c_z - K_h c_x H_x = E$       at $z = -H.$  (12)

In Eq. (10), $w_s$ is the settling velocity and $K_h$ and $K_v$ are the horizontal and vertical eddy diffusivity. Boundary condition (12) is valid only when assuming $H_x$ is much smaller than unity. We assume that $K_v$ is related to the vertical eddy viscosity coefficient $A_\nu$ as $K_v = A_\nu/\sigma_\rho$, where $\sigma_\rho$ is the Prandtl-Schmidt number that converts viscosity to diffusivity. The erosion

flux $E$ is related to the so-called reference concentration $c_\star$ through $E = w_s c_\star$. In turn, the reference concentration is defined as

$$c_\star(x,t) = \rho_s \frac{|\tau_b(x,t)|}{\rho_0 g' d_s} a(x),$$  (13)

where $\rho_s$ is the density of sediment, $\tau_b(x,t) = \rho_0 A_v u_z$ is the bed shear stress (again assuming $H_x \ll 1$), $g' = g(\rho_s - \rho_0)/\rho_0$ is the reduced gravity, $d_s$ is the mean grain size, and $a(x)$ is the availability of easily erodible fine sediment.

The dimensionless sediment availability function $a(x)$ is unknown and can be determined by imposing the so-called *morphodynamic equilibrium condition*. Following Friedrichs et al. (1998); Huijts et al. (2006); Chernetsky et al. (2010), we assume that the total amount of sediment in the estuary varies on a time scale that is much longer than that at which the easily erodible sediment is redistributed. In that case, the availability of sediment can be determined by assuming that the tidally averaged transport of sediment is divergence free, i.e. there is a balance between the tidally averaged erosion

and deposition at the bottom $z = -H(x)$. This condition also provides the horizontal boundary conditions to the sediment dynamics equation, stating that there is no net sediment transport across the landward boundary at $x = L$. The resulting morphodynamic equilibrium condition can be written as (Chernetsky et al., 2010)

$$B \left\langle \int_{-H}^{\zeta} (uc - K_h c_x) dz \right\rangle = 0.$$  (14)

As the concentration $c$ in Eq (14) depends on the availability $a(x)$, the above condition is an equation for $a$. This determines

the availability up to a constant factor $a^*$ that should be prescribed on input. This factor determines the total amount of sediment in the system. As the amount of sediment in the system directly affects the concentration in the water column, the absolute magnitude of the concentration may be calibrated directly by changing $a^*$. The relevant result of the sediment model therefore consists of the relative differences between concentrations at different locations along the estuary instead of the absolute magnitude of the concentration.





### 4.2 Scaling

The first step in the perturbation approach is the scaling of the equations. This approach uses a systematic mathematical procedure to determine the relative importance of the different terms in the equations for water motion and sediment dynamics. The most dominant terms will be called *leading-order terms*. Terms that are significantly smaller than these

leading-order terms will be further categorised according to their relative importance. The most dominant terms, after separating leading-order terms, are called *first-order terms*. This categorisation continues, with all terms of second or higher order generally referred to here as *higher-order terms*.

The scaling requires four crucial assumptions. Firstly we assume

$$\varepsilon = \frac{\zeta}{H} \ll 1, \tag{15}$$

i.e. the ratio of the typical water level amplitude to the depth is much smaller than unity. The small parameter $\varepsilon$ is used to define of which order a term is. A term is defined to be of first order if its typical relative magnitude is of order $\varepsilon$ compared to the leading order terms. Similarly, an $n^{\text{th}}$-order term is of order $\varepsilon^n$ with respect to the leading order terms.

Secondly, it is assumed that the typical tidal wave length and the typical length-scale of bathymetric variations are of the same order of magnitude as the length of tidal influence into the estuary. This implies that sudden local bathymetric

variations are not allowed. Rather, bathymetric changes should be smooth over the length of the estuary. Likewise, the method is restricted to long waves, such as tides. Short waves, such as wind waves, are not accounted for. As a consequence of this assumption, the non-linear advection term $uu_x + wu_z$ in Equation (1) and $uc_x + wc_z$ in Eq (10) scale with $\varepsilon$. It is found that, by these two assumptions, the leading-order terms are all linear, while all non-linearities in the velocity, concentration and water level elevation only appear as first-order or higher-order effects.

Thirdly, it is assumed that the horizontal density gradient is small. More precisely, the internal Froude number should be of order $\varepsilon$ or equivalently $\rho_x L_{\text{tide}}/\rho_0$ should be of order $\varepsilon^2$, where $L_{\text{tide}}$ is the length of tidal influence. As a consequence, the baroclinic pressure term $g \int_z^{R+\zeta} \frac{\rho_x}{\rho_0} d\tilde{z}$ in Equation 1 is of order $\varepsilon$. Finally, the horizontal diffusion term $(K_h c_x)_x$ is assumed to be of order $\varepsilon^2$.

### 4.3 Perturbation approach

Instead of neglecting higher-order non-linear effects, as is done in conventional linearisation techniques, the perturbation approach expands these non-linearities into a series of linear estimates. To this end, the solution variables $u$, $w$, $\zeta$ an $c$ are written as an asymptotic series ordered in the small parameter $\varepsilon$, i.e.

$$u = u^0 + u^1 + u^2 + \dots,$$
$$w = w^0 + w^1 + w^2 + \dots,$$

$$\zeta = \zeta^0 + \zeta^1 + \zeta^2 + \dots,$$
$$c = c^0 + c^1 + c^2 + \dots,$$




where $[\cdot]^0$ denotes a quantity at leading order, $[\cdot]^1$ denotes a quantity of order $\varepsilon$, $[\cdot]^2$ of order $\varepsilon^2$ etcetera. Additionally the eddy viscosity and diffusivity, density, tidal forcing, river discharge and fall velocity are written as similar series. These series are substituted into the equations. The resulting equations are still equivalent to the original system of equations. The analysis up to this point has merely identified what terms in the equations are of leading and higher orders.

The perturbation approach is illustrated here for the hydrodynamic equations. A first approximation of the equations for the hydrodynamics can be made by neglecting all terms of higher orders. The remainder is a linear set of leading-order equations in $u^0$, $w^0$ and $\zeta^0$. The water motion is forced by the leading-order tidal forcing $A^0$ and an optional leading-order river discharge $Q^0$ (numerical solution method only). If the eddy viscosity is assumed to be independent of the flow, this equation describes the linear propagation of the tide and river flow through the estuary without considering tide-river

interactions or density variations.

    An improved approximation of the total solution results from constructing the balance of first-order terms. This consists of a linear set of equations in $u^1$, $w^1$ and $\zeta^1$, forced by the first-order tidal forcing $A^1$, first-order river discharge $Q^1$ and the linear estimates of the non-linearities acting on the leading-order flow. These non-linearities include the effects of momentum advection, the tidal return flow and velocity-depth asymmetry. The tidal return flow is the flow that com-

pensates for the mass transport due to correlations between the tidal velocity and surface variation. The velocity-depth asymmetry accounts for the effect that the velocity profile differs between ebb and flood due to different water levels. Since the baroclinic pressure term is assumed to be of first order, the baroclinic pressure is an additional forcing to the first-order system. Finally, time-variations of the eddy viscosity may be described at first-order, so that the interactions between these time variations and the leading-order flow appear as a forcing at the first order. Because of the linearity

of the equations, all forcing terms to the first-order equations, summarised in Table 2, can be evaluated separately. The total first-order solution is obtained by adding the solutions of each forcing term.

    For the sediment balance, the leading-order equation describes a local balance between vertical turbulent mixing and the settling of sediment. The balance is only forced at the bottom where sediment is locally resuspended. The first-order equation describes a similar balance between vertical diffusion and the settling of sediment. This equation is

forced by local first-order resuspension and by linear estimates of non-linearities acting on the leading order sediment concentration, including the effects of sediment advection, also known as spatial settling lag effects (Table 2). Another non-linear term is the interaction between the sediment concentration and the surface elevation. If the eddy diffusivity and fall velocity have first-order contributions, their correlation with the leading-order concentration appear as first-order effects as well. Similar to the hydrodynamics, all the contributions to the sediment concentration by different forcing terms

can be evaluated separately.

    Similar to the approach outlined above for the first-order terms, higher-order approximations of both the hydrodynamics and sediment dynamics can be made by composing a balance of the terms on second, third and higher orders. It is assumed that all external forcing terms act on either the leading or the first order. The second and higher orders therefore only contain estimates of non-linear interactions of lower order contributions. The sum of all estimates of the non-linear

terms at all orders should return the total solution to the original non-linear system of equations. If the scaling assumptions

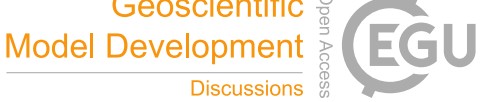



are satisfied, it follows that the contributions at higher order rapidly become smaller. The solutions at leading and first order therefore already provide a fairly accurate estimate of the total solution. The higher-order systems are nevertheless useful in cases where the scaling assumptions are only marginally satisfied or when studying a particular process that involves a non-linear interaction that appears at higher order.

## 4.4  Harmonic decomposition

Since the set of leading-order and first-order equations is only forced by a sub-tidal flow, the $M_2$ tide (i.e. the tidal constituents with a period of 12.42 hours) and its overtides, the solution also consists of these frequency components. Using this knowledge, we will write the solutions as a sum of these tidal components. This approach has major advantages, because it removes the explicit time dependency and time derivatives and therefore removes the need to solve the equations by time-stepping.

The harmonic decomposition is exact under the simplifying assumptions that the leading-order eddy viscosity, eddy diffusivity, partial slip parameter and fall velocity are described by a sub-tidal component only. The solution of the hydrodynamics at every order then has a restricted number of harmonic components. In the common case where the leading order is forced by an $M_2$ tide and the first order is forced by a sub-tidal and $M_4$ component, the solution only contains these three components up to the first order. For the sediment dynamics, infinitely many harmonic components exists, but the same three components provide a reasonably accurate estimate of the concentration up to the first order.

If time variations of the leading-order eddy viscosity, eddy diffusivity, partial slip parameter or fall velocity are allowed, the solution of both the hydrodynamics and sediment dynamics consists of an infinite number of tidal components at every order. The tidal signal therefore needs to be approximated by a limited number of tidal components. The harmonic decomposition then forms an approximation, the accuracy of which can be made arbitrarily high by including more tidal components.

## 4.5  Semi-analytical versus numerical method and analysis of the sediment transport

The iFlow standard modules for hydrodynamics and sediment dynamics include two solution methods: a numerical and semi-analytical method. The semi-analytical solution method follows Chernetsky et al. (2010). The model uses fully analytical formulations for the vertical velocity and sediment profiles, but uses a numerical method to solve for the water level elevation. The semi-analytical method is fast, has a guaranteed high accuracy and is directly comparable to classical theory on e.g. tidal propagation. The method however poses restrictions on the external model forcing. These restrictions include the requirement that the eddy viscosity, eddy diffusivity and fall velocity are constant in time up to leading order and uniform over the depth and that the partial slip parameter is constant in time as well. The semi-analytical model is forced by a leading-order $M_2$ tide and first-order $M_4$ tide and river discharge. An advantage of these restrictions is that they lead to a concise set of resulting frequency components, which can be easily analysed. The leading-order water motion only contains an $M_2$ component forced by an externally prescribed tidal amplitude. The first-order water motion is described by a sub-tidal and $M_4$ component forced by the external tide and river discharge as well as by linear estimates





of non-linear terms and the baroclinic pressure. Conversely, the leading-order sediment dynamics is described by sub-tidal, $M_4$, $M_8$ etc. components due to tidal resuspension. The first-order is described by $M_2$, $M_6$, $M_{10}$ components due to tidal resuspension, sediment advection and other non-linearities. In the semi-analytical model, only the sub-tidal and $M_4$ components of the leading-order concentration and the $M_2$ component of the first-order concentration are considered.

This is because these are the dominant components for the transport and trapping of sediment.

Within the semi-analytical approach, the morphodynamic equilibrium condition Eq (14) is a sub-tidal balance of sediment transport terms at second order. We can distinguish between three types of transport terms. The first describes the covariance between the velocity and concentration, i.e. $\langle \int_{-H}^{0} uc\,dz \rangle$. The dominant covariance terms that result in a sub-tidal transport are $\langle \int_{-H}^{0} u^0 c^1\,dz \rangle$ and $\langle \int_{-H}^{0} u^1 c^0\,dz \rangle$. The term $u^0 c^1$ only generates a sub-tidal transport due to the

covariance between the leading-order $M_2$ flow and $M_2$ variation of the first-order concentration. The term $u^1 c^0$ generates transport due to $M_4$-$M_4$ covariance and the product of both sub-tidal contributions. As the model computes the effect of different physical mechanisms contributing to $u^1$ and $c^1$ (see Table 2), the transport terms can be subdivided further into the transport caused by particular physical mechanisms. This way, we obtain a subdivision of $\langle \int_{-H}^{0} u^1 c^0\,dz \rangle$, with components named after the different contributions to $u^1$. Likewise, the components in the subdivision of $\langle \int_{-H}^{0} u^0 c^1\,dz \rangle$

are named after the contributions to $c^1$. One exception to this is the 'erosion' contribution to $c^1$, which is again subdivided further into the $u^1$ velocity contributions that cause the erosion. Additional to these terms, the model includes the subtidal transport by $\langle \int_{-H}^{0} u^1_{\text{river}} c^2_{\text{river-river}}\,dz \rangle$, i.e. the covariance between the river-induced velocity and the river-induced sediment resuspension. Even though this transport is a fourth-order term according to the scaling, it typically becomes the dominant term near the end of the tidal influence.

The second type of transport term is the covariance between the velocity, concentration and the varying water surface elevation, with dominant contribution $u^0 c^0 \zeta^0$. No further subdivision of this term can be made. This term represents the drift of sediment with the moving surface and is largely compensated for by the tidal return flow, which is part of $\langle \int_{-H}^{0} uc\,dz \rangle$. Therefore we will consider the transport due to this drift and the tidal return flow together as one term under the name tidal return flow.

The final type of transport terms are the terms involving the horizontal eddy diffusivity, $\langle K_H c^0 \rangle$ and $\langle K_H c^2_{\text{river-river}} \rangle$. It is assumed that the horizontal diffusivity is constant in time, so that the term $K_h c^1$ is zero averaged over the tide. The diffusive transport thus describes horizontal background diffusion of the tide- and river-induced resuspended sediment. Physically, this background diffusion is caused by unresolved flows, such as lateral circulation.

The numerical method was introduced, because the assumptions on the forcing in the semi-analytical method can be

too restrictive for specific applications. The numerical approach allows for a time-varying and depth-varying eddy viscosity, eddy diffusivity and fall velocity and a time-varying partial slip parameter. The model may be forced by the $M_2$ tide and any over-tidal component at both the leading and the first order and allows for either a leading-order or first-order river discharge. This means that the numerical model may be used with the same restrictions as the semi-analytical method, but these restrictions may be relaxed for further functionality. An overview of the differences between the restrictions in

the semi-analytical and numerical methods is provided in Table 3

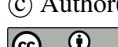



Some of the additional functionality of the numerical method affect the sediment transport balance. The possible addition of more harmonic components leads to additional transport terms, such as the the transport due to the $M_6$-$M_6$ covariance between the velocity and concentration. When a sub-tidal or $M_4$ velocity is entered at the leading-order, e.g. through the river discharge or externally prescribed tide, the covariance between the leading-order velocity and concentration, $\langle \int_{-H}^0 u^0 c^0 \, dz \rangle$, yields a sub-tidal contribution. According to the scaling, this contribution dominates over the transport contributions mentioned earlier, so that those contributions should no longer be considered. The term $\langle \int_{-H}^0 u^0 c^0 \, dz \rangle$ in the new balance can again be subdivided according to the physical mechanisms that contribute to the velocity and concentration. The balance now only concerns the leading-order velocity and concentration, for which the model computes only one or two contributions (see Table 2). The subdivision of the transport therefore leads to much fewer terms and typically provides less insight into the underlying physics.

The choice to keep a simulation within the restrictions of the semi-analytical method or extend it to the full possibilities of the numerical method thus has a direct effect on the ability to analyse the results. This is an example of the classical trade-off between model complexity and ability to analyse the results as was mentioned in the introduction. A major strength of iFlow is that it offers one software environment where one can experiment with the degree of complexity required for a simulation for a specific application.

## 5 Introduction to the modules for turbulence and salinity

### 5.1 Turbulence models

iFlow provides a number of modules to parametrise the eddy viscosity and roughness parameter (see also Table 1), referred to as the turbulence model. The simplest turbulence model available is implemented in the module 'Turbulence-Uniform' and assumes a vertically uniform eddy viscosity and constant partial-slip roughness parameter, which may only vary with the depth (see also Friedrichs and Hamrick (1996); Schramkowski and De Swart (2002)) according to

$$A_\nu = A_{\nu 0} \left( \frac{H + R}{H(x = 0)} \right)^m ,$$

(16)

$$s_f = s_{f,0} \left( \frac{H + R}{H(x = 0)} \right)^n ,$$

(17)

with $A_{\nu 0}$, $s_{f,0}$, $m$ and $n$ provided as input to the model. The input parameters $A_{\nu 0}$ and $s_{f,0}$ may include time-variations (in combination with the numerical hydrodynamics only).

The second turbulence model implemented in the module 'TurbulenceParabolic' is similar, but assumes the eddy viscosity to have a parabolic profile in the vertical direction. This turbulence model assumes $s_f \to \infty$, so that the bottom boundary condition for the hydrodynamics reduces to a no-slip law. The roughness is instead described by a roughness




height $z_0$. The formulations for $A_\nu$ and $z_0$ read

$$A_\nu = A_{\nu 0} \left( \frac{H+R}{H(x=0)} \right)^m (z_s^* - z^*)(z_0^* + z^* + 1), \tag{18}$$

$$z_0 = z_0^* \left( \frac{H+R}{H(x=0)} \right)^{n+1}. \tag{19}$$

The parameters $A_{\nu 0}$ and $z_0^* = z_0(x=0)/H(x=0)$, $m$ and $n$ are provided as input, $z^* = z/(H+R)$ and the dimension-

less surface roughness $z_s^*$ is determined by the model such that $A_\nu$ equals $10^{-6}$ m$^2$/s at the surface, i.e. approximately the molecular viscosity.

Finally, iFlow includes a set of modules named 'KEFitted'. These turbulence modules define parametrisations for $A_\nu$ and $s_f$ derived by fitting the results of iFlow to the results of a one-dimensional numerical model with $k - \varepsilon$ closure (see e.g. Dijkstra et al. (2016b)) for a large number of barotropic tidal model configurations. The turbulence closures provide a

number of options. The most important option is the choice of roughness parameter to provide on input. If the roughness parameter $s_{f,0}$ is provided, the turbulence model uses the relation

$$A_{\nu 0} = 0.5 s_f (H + R + \zeta), \tag{20}$$

$$s_f = s_{f,0} \left( \frac{H+R}{H(x=0)} \right)^n, \tag{21}$$

This model only has the calibration parameters $s_{f,0}$ and $n$ and thus eliminates the need to calibrate $A_{\nu 0}$ and $m$.

Alternatively, the turbulence model may be provided with a roughness parameter $z_{00}^*$ The formulations for the eddy viscosity and partial-slip roughness then read

$$A_\nu = \alpha u^* (H + R + \zeta) f_1(z_0^*), \tag{22}$$

$$s_f = \beta u^* f_2(z_0^*), \tag{23}$$

$$z_0^* = z_{00}^* \left( \frac{H+R}{H(x=0)} \right)^n. \tag{24}$$

where $z_{00}^*$ is provided as input and $\alpha$, $\beta$, $f_1(z_0^*)$ and $f_2(z_0^*)$ are known parameters and functions (see the manual for details). These formulations relate the vertically uniform eddy viscosity and partial-slip parameter to the local bed shear stress velocity and water depth. This introduces non-linearity as the flow velocity and the water surface elevation naturally depend on the eddy viscosity and partial-slip parameter. This non-linearity is resolved by an iteration loop over the turbulence and hydrodynamic modules as exemplified in Section 2.

iFlow implements four modules that implement the above 'KEFitted' relations. The modules 'KEFittedLead', 'KEFittedFirst' and 'KEFittedHigher' make an ordering of the above equations to determine the leading-order, first-order and higher-order eddy viscosity and partial-slip parameter. The module 'KEFittedTruncated' uses the sum of all computed orders of the velocity and water surface elevation to compute a total eddy viscosity and roughness parameter without ordering (i.e. a truncation method).





## 5.2 Salinity

The iFlow standard modules include two types of salinity models: diagnostic (i.e. prescribed) and prognostic (i.e. resolved). The diagnostic modules prescribe a sub-tidal vertically uniform (well-mixed) salinity that varies in the along-channel direction. The module 'SalinityHyperbolicTangent' formulates this as (see also Warner et al. (2005); Talke et al. (2009))

$$s = \frac{s_{\text{sea}}}{2} \left( 1 - \tanh\left( \frac{x - x_c}{x_L} \right) \right) \tag{25}$$

and 'SalinityExponential' formulates this as

$$s = s_{\text{sea}} \exp\left( -\frac{x}{L_s} \right). \tag{26}$$

The prognostic salinity model follows work done by McCarthy (1993); Wei et al. (2016). The model is based on the perturbation approach, where it is assumed that the leading-order salinity consists of a sub-tidal vertically uniform (well-mixed) salinity. Vertical and temporal variations of the salinity appear at higher orders. For more information we refer to Wei et al. (2016).

## 5.3 Reference level

The hydrodynamic module relies on the water depth being positive and much larger than the time varying surface elevation. The model fails or becomes inaccurate if the bottom lies above or close to MSL. In many cases this problem can be resolved by the iFlow ReferenceLevel module. This module computes a quick estimate of the sub-tidal water level elevation based on the river-induced set-up. This is often sufficient, because the river is often the dominant flow term in the most upstream reach, where the bottom level is highest.

The river-induced set-up is estimated numerically by from the leading-order momentum and depth-averaged continuity equations, assuming it is purely forced by a constant discharge $Q$ and the resulting water level elevation is given by $R$. These equations read

$$-gR_x + (A_\nu u_z)_z = 0, \tag{27}$$

$$\left( B \int_{-H}^{R} u \, dz \right) = Q. \tag{28}$$

This system is non-linear in $R$ as the integral in the second equation contains $R$ in the integration boundary and $u$, which depends on $R$ according to the first equation. Nevertheless, the system can be solved without iterating by starting at the mouth and working upstream. At the mouth ($x = 0$), $R = 0$ by definition. Therefore $R_x$ can be computed from the above system of equations. The value of $R$ at the next grid point $x = \Delta x$ follows from a simple first-order routine: $R(\Delta x) = R(0) + R_x(0)\Delta x$. The total reference level follows by repeating this procedure for all horizontal grid cells. More accurate





computations of the river-induced set-up follow from the hydrodynamic modules, so that the relatively low numerical accuracy of the reference level computation will not reduce the precision of the overall result.

The reference level still depends on the eddy viscosity. If a 'KEFitted' turbulence model is used, the eddy viscosity in turn depends on the reference level. To resolve this interdependency efficiently, without needing to iterate between the
turbulence model and reference level module, the 'KEFitted' turbulence models have a built-in reference level routine. The module ReferenceLevel therefore does not have to be used if 'KEFitted is used.

### 5.4 Sensitivity module

iFlow's standard sensitivity module 'Sensitivity' provides a powerful analysis tool. The module allows to vary the values of several input variables. The list of variables to vary over needs to be provided on input as a list with one or more variable
names. The next input parameters are lists with all values for each of these variables. The final input parameter indicates whether all combinations of parameter values should be tested or whether the values of all variables should be changed simultaneously. The sensitivity analysis is therefore a general tool that may be combined with any set of modules to loop over any set of variables and values.

The sensitivity module is an iterative module, which forces the iFlow core to construct an iteration loop. In the first
iteration, the sensitivity module only requires its parameters from the input file. The iFlow core could therefore place the sensitivity analysis anywhere in the call stack. However, one of the criteria for the core is to make iteration loops as short as possible. It therefore places the sensitivity module as late in the call stack as possible, but before the first module that needs one of the variables in the sensitivity study. The sensitivity analysis simply takes the first values from the lists on input and assigns them to their respective variables. The output of the sensitivity module thus consists of the variables
looped over. This is not known a-priori and cannot be hard-coded in the registry. It is therefore determined dynamically, based on the input.

For consecutive iterations the sensitivity module requires all variables that are needed on output. This ensures that the output is saved before starting the next iteration. During each iteration, the sensitivity module takes the new values from input and assigns them to the variables to loop over. The loop is stopped automatically when there are no more values to
loop over. An example of the use of the sensitivity module is given in the model evaluation in Section 6.2.

### 6 Model evaluation

The use of a 2DV perturbation approach for hydrodynamics or sediment dynamics similar to iFlow's semi-analytical method has been demonstrated before by e.g. Chernetsky et al. (2010); Wei et al. (2016). An application of iFlow itself has been presented before by Dijkstra et al. (Manuscript submitted to JGR-Oceans). They use the time-dependent leading-
order eddy viscosity to compute the effect of large eddy viscosity variations on exchange flows in estuaries. This is used to make a thorough decomposition of the physical contributions to this exchange flow.



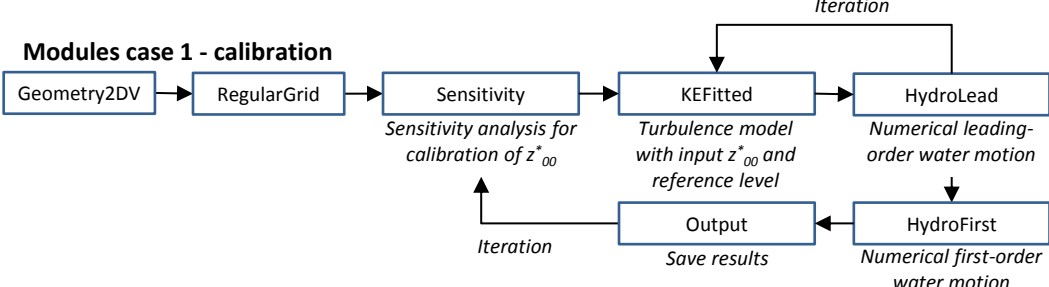

(a) Modules used in the calibration stage of the Yangtze case study.

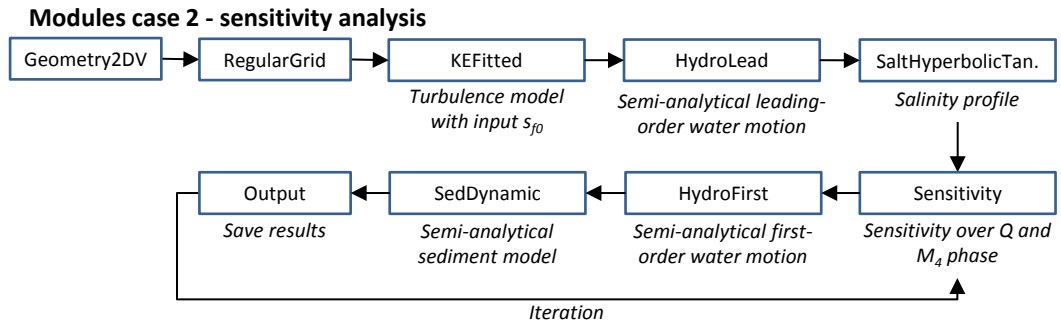

(b) Modules used in the sensitivity study in the Scheldt case study.

**Figure 4.** Modules used in two stages of the case studies, set into the correct order and including iteration loops over groups of modules.

Here, iFlow is applied to two case studies. The aim of these cases is to show the application of iFlow, demonstrate ways it can be used to analyse the results and qualitatively compare the model results to measurements. While this aim requires discussing some of the physical mechanisms observed in the model, these physical mechanisms are not the focus of this section. The first case study is an assessment of hydrodynamic interactions between the tidal and river-induced flow in the tidal part of the Yangtze River, China. This case demonstrates some of the advanced hydrodynamic settings in iFlow using the numerical solution procedure, including the use of the reference level, leading-order river discharge and velocity-dependent eddy viscosity. The model calibration is also demonstrated. The second case study presents an assessment of the estuarine turbidity maximum (ETM) in the Scheldt River estuary. This case will use the semi-analytical modules for hydrodynamics and sediment dynamics. This case also features a demonstration of the sensitivity analysis module of iFlow.

As a result of iFlow's flexible modular structure, the modules used are different from application to application. The modules used in the two applications presented below are shown in Figure 4. Both cases use the modules for generating





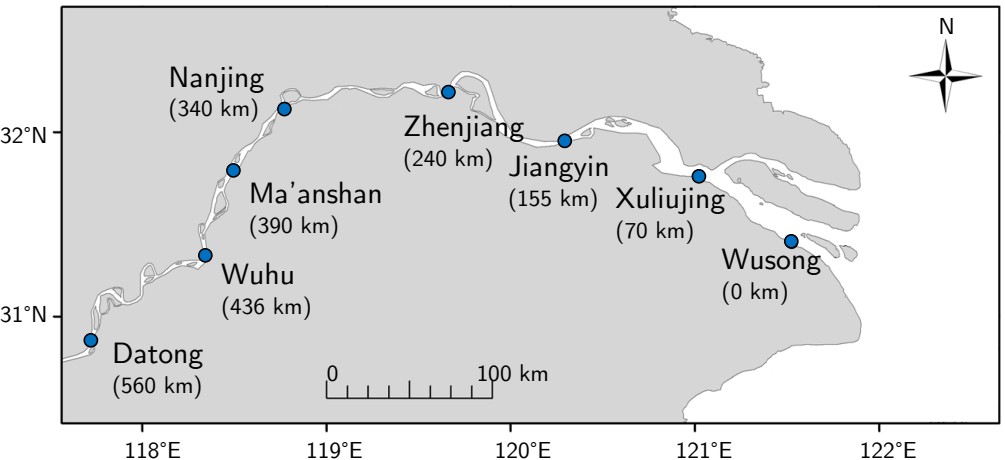

**Figure 5.** The tidal Yangtze River. Adapted from Guo et al. (2016)

the model domain and grid. The first case then makes use of a module for the reference level and velocity-dependent eddy viscosity and partial slip parameter. As this module requires the leading-order velocity, it automatically iterates over the leading-order hydrodynamics module until the result has converged. The calibration routine calibrates on both the leading-order and first-order hydrodynamics using the roughness parameter in the turbulence model. It therefore automatically

constructs an iteration loop over these modules. The second case is a linear sequence of modules without any need for iteration loops. Only the sensitivity analysis initiates a loop. This loop is kept as small as possible, so that a loop over the discharge and the externally prescribed $M_4$ tidal phase only requires a loop over the first-order hydrodynamics and sediment dynamics.

### 6.1   Tide-river interactions in the tidal Yangtze River

The tidal part of the Yangtze river in China stretches from its mouth near Shanghai approximately to Datong, 560 km upstream, where the tidal influence is typically negligible, see Figure 5. In the model schematisation we locate the mouth at the station of Wusong, in the south branch of the estuary, where the river forms a single-channel system. The effect of the North Branch is neglected. The domain is then 560 km long from Wusong to Datong. In order to ensure that the tidal wave damps out, the model domain is extended to 1500 km, of which only the first 560 km are analysed.

Measurements of the width-averaged bed level and near-surface width are provided by Guo et al. (2014). The bed level is characterised by large variations caused by local width variations and river bends. Smoothing this profile, the bed level is well characterised by a horizontal bed with a depth of 10 m. The width is strongly converging from 25 km at the mouth to a fairly constant 3 km between 200 and 500 km. This width profile is approximated by the exponent of a rational



function given by

$$B = 1000 \exp \left( \frac{3.8 \cdot 10^{-5} x^2 + 10}{8.8 \cdot 10^{-11} + 2.5 \cdot 10-5 + 3.2} \right).$$

We will distinguish between two forcing conditions: wet and dry season conditions. For both we will assume average tidal conditions, for which the primary forcing components are a leading-order $M_2$ tide with amplitude 1.09 m and a

first-order $M_4$ tide with amplitude 0.22 m and a phase difference of $-44$ degrees (Guo et al., 2016). We assume a representative discharge of 50,000 m³/s for the wet season and 15,000 m³/s for the dry season. In both conditions, the river is assumed to force the water motion at leading order. The effects of salinity or sediment on the flow are not considered for simplicity. We will only consider the leading-order eddy viscosity computed by the turbulence module 'KEFittedLead' with roughness parameter $z_{00}^*$ (see Eq (22)-(24)). The leading-order eddy viscosity is assumed uniform in

the vertical and the leading-order eddy viscosity and partial slip parameter are assumed constant in time and dependent on the leading-order velocity. The eddy viscosity and partial slip roughness parameter are therefore a function of the leading-order $M_2$ tide and the river discharge.

The model is calibrated by tuning the roughness parameter $z_{00}^*$ to measured water levels for the wet season. The model is calibrated through the sensitivity analysis module. This module constructs a loop over the hydrodynamic modules for a

range different $z_{00}^*$ values. The results of each computation are compared to the measurements by using the cost function introduced by Jones and Davies (1996). The result is plotted in Figure 6, which shows the value of the cost function for the $M_2$ tide and $M_4$ tide as a function of $z_{00}^*$. The actual value of the cost function is not displayed, since there is no interpretation to this value. The best fit to the measurement is found for the smallest cost, which is for $z_{00}^* = 9.6 \cdot 10^{-5}$ for the $M_2$ tide and $z_{00}^* = 1.3 \cdot 10^{-4}$ for the $M_4$ tide. Only one value for $z_{00}^*$ can be chosen. Since these values are close

together and we proceed with a rounded value of $z_{00}^* = 1 \cdot 10^{-4}$. The same roughness value is used for the dry season case.

The resulting water level amplitude and phase are plotted in Figure 7. The lines show the model results for the sub-tidal flow, $M_2$ and $M_4$ tides in the wet season (dashed line) and dry season (solid line). The dots and crosses indicate measurements presented by Guo et al. (2016) for the dry and wet seasons respectively. We find a good correspondence

between the measured tidal water level amplitude and phase. This is even true for the dry case, for which the model has not been recalibrated. We additionally find a good correspondence between the measured and modelled sub-tidal water levels, even though the model has also not been calibrated for these. Most importantly, the model captures the correct trends between the wet and dry season, such as increased tidal damping of the $M_2$ and $M_4$ tide in the wet season and a slower propagation (i.e. larger phase differences) of the $M_2$ tide in the wet season.

We will demonstrate how the model can be used to uncover the main processes that cause the differences in tidal propagation between the dry and the wet season. The non-linear first-order processes that involve tide-river interactions are advection, tidal return flow and velocity-depth asymmetry. Through the perturbation method, these processes can be inferred directly from the model results. The correction to the $M_2$ tidal amplitude from each of these terms is only of the



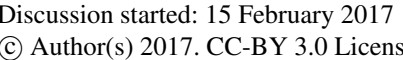



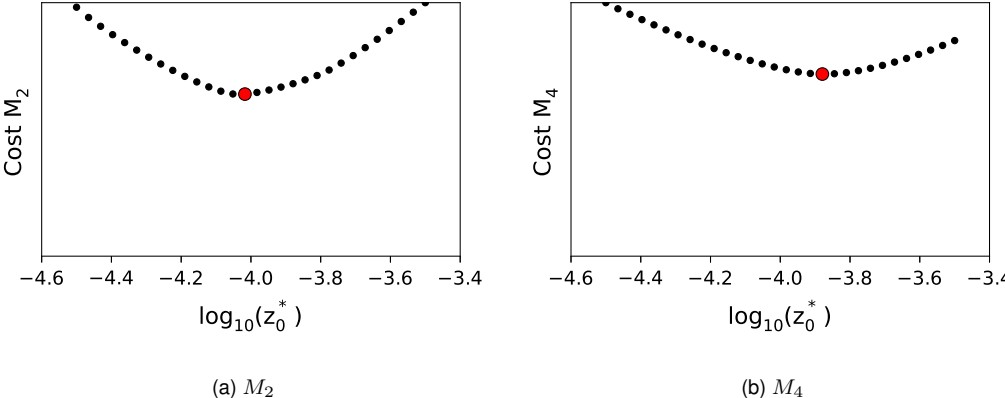

(a) $M_2$                    (b) $M_4$

**Figure 6.** Cost function that measures the error between the measurements of the water level amplitude and model results for a range of values of the calibration parameter $z_{00}^*$. The error is plotted for the $M_2$ tide (a) and the $M_4$ tide (b). The red dot marks the minimum error. The absolute value of the error has no interpretation, therefore no values are shown on the vertical axis.

order of some centimetres in both the dry and the wet season. The higher-order non-linear terms (not shown here) pose even smaller corrections to the $M_2$ tide.

The model is rerun without the reference level module (i.e. $R = 0$) to see the effect of the difference in reference depth between the dry and the wet season. The resulting water level amplitude is plotted in Figure 8. The first striking observation is that the sub-tidal water levels are now unrealistically high in the wet season ($> 100$ m). Since the river-induced water level set-up is large, the assumption that $\varepsilon \ll 1$ (i.e. the ratio of the water level and depth is small) is violated, leading to an unrealiable computation. Nevertheless, the $M_2$ and $M_4$ tide are hardly affected by this and show the same characteristics of the tide-river interactions as before. We can thus conclude that the reference level is an essential model feature in model cases with a large river-induced set-up, but does not seem to be essential in tide-river interactions.

The effect of the river flow on the eddy viscosity and partial slip parameter can be assessed switching off the dependency of these quantities on the river flow. The 'KEFitted' turbulence module provides an option to switch off any physical mechanism that can be separated explicitly from the solutions. The river flow can therefore be disregarded in the computation of the eddy viscosity and partial slip parameter, while the dependence on the $M_2$ tidal velocity is still accounted for. The resulting water level amplitude is plotted in Figure 9. The $M_2$ and $M_4$ tides now do not experience a sufficient degree of damping to vanish at the 560 km point. Also, the differences in the $M_2$ tidal amplitude between the dry and wet season have nearly vanished. The main effect of the river run-off on the tidal amplitude is thus through the effect the river flow has on the eddy viscosity and partial slip parameter.





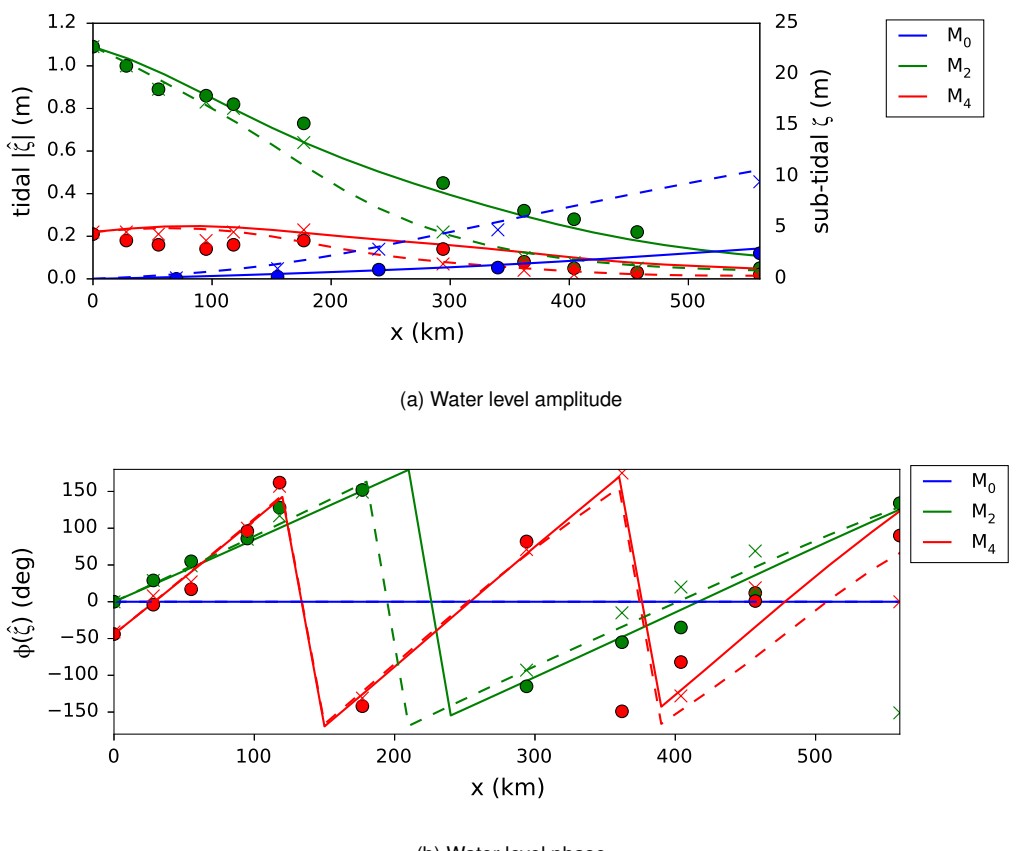

(a) Water level amplitude

(b) Water level phase

**Figure 7.** Water level amplitude (a) and phase (b) for the Yangtze case in a dry season situation (solid line and dots) and a wet season situation (dashed line and crosses). The lines represent the sub-tidal (blue), $M_2$ (green) and $M_4$ (red) model results. The dots and crosses are measurement data presented by Guo et al. (2016).

## 6.2 ETM location in the Scheldt River estuary

The tidal Scheldt river, situated in the southwest of the Netherlands and northwest of Belgium, runs from its mouth in the North Sea to a set of locks and sluices near Ghent, 160 km upstream. The river serves as the main shipping channel to the port of Antwerp, which is located at approximately 90 km upstream from the mouth. Dredging activities for maintaining and deepening the shipping channel are one of the main reasons to study the development of the fine sediment dynamics in the Scheldt River. The estuary is over 6 km wide and averages a depth of 15 m at its mouth. The estuary converges to a width of about 50 m and an average depth of about 3 m at the end of the tidal influence. To obtain a schematized depth and width profile, a polynomial is fitted through the geometry data of 2013 (Coen et al., 2015). The depth profile is





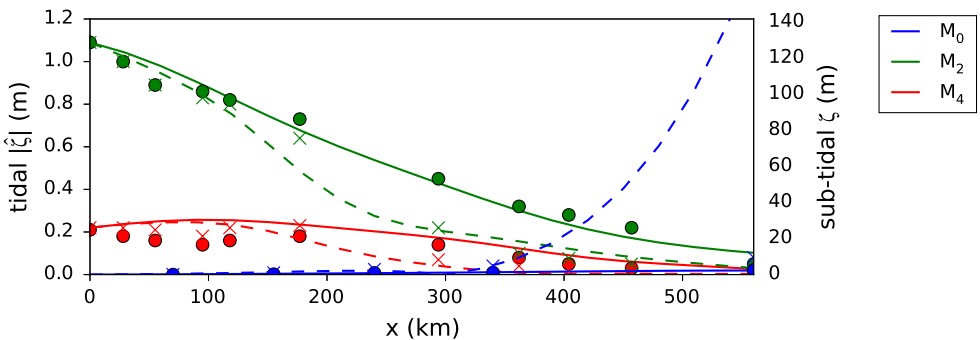

**Figure 8.** Water level amplitude for the Yangtze case when the computation of the reference level is omitted. The figure shows results for a dry season situation (solid line and dots) and a wet season situation (dashed line and crosses), see Figure 7 for more explanation. Omitting the reference level leads to unrealistically high modelled subtidal water levels in the wet season.

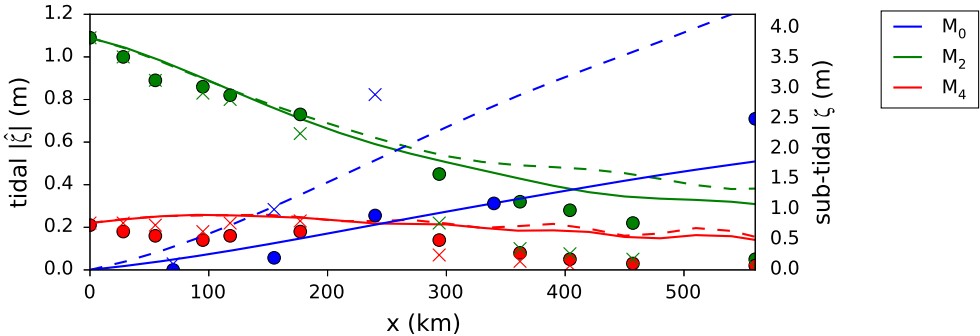

**Figure 9.** Water level amplitude for the Yangtze case when the effect of the river discharge on the eddy viscosity and partial slip parameter is omitted. The figure shows results for a dry season situation (solid line and dots) and a wet season situation (dashed line and crosses), see Figure 7 for more explanation. The $M_2$ tide is damped less and there is a smaller difference between the $M_2$ tidal amplitude in the wet and dry seasons.



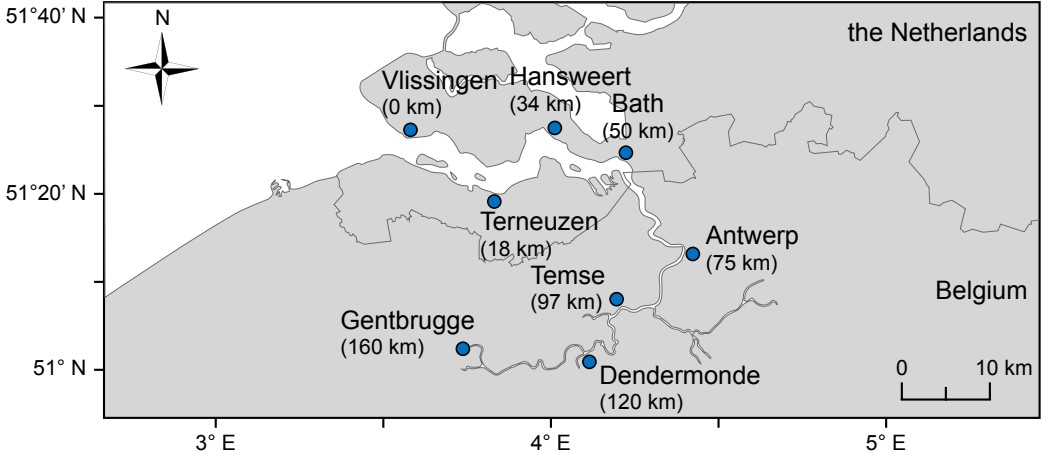

**Figure 10.** The Scheldt River estuary

approximated by a fifth-order polynomial given by

$$H(x) = -2.9 \cdot 10^{-24} x^5 + 1.4 \cdot 10^{-18} x^4 - 2.4 \cdot 10^{-13} x^3$$
$$+ 1.7 \cdot 10^{-8} x^2 - 5.2 \cdot 10^{-4} x + 15.3,$$

and the width profile is approximated by an exponent of a rational function given by

$$B(x) = \exp\left(\frac{-0.027 \cdot 10^{-3} x + 1.9}{5.0 \cdot 10^{-11} x^2 - 9.2 \cdot 10^{-6} + 1}\right).$$

The model is forced by a leading-order $M_2$ tidal amplitude of 1.77 m and a first-order $M_4$ tidal amplitude of 0.14 m, which is -1.3 degrees out of phase with the $M_2$ tide. The eddy viscosity $A_v$ is computed using the 'KEFittedLead' module using $s_{f,0}$ as the input parameter and using $n = 0$ (see Eq (20)-(21)). Therefore the partial slip parameter is constant in space and time and the eddy viscosity is assumed to be uniform over the vertical, linearly dependent on the depth and

constant in time. The salt water influence typically reaches up to the port of Antwerp (i.e. 90 km). It is assumed that the salinity is well-mixed and can be described by a prescribed horizontal salinity profile, which is obtained by fitting a tangent hyperbolic function to summer and winter measurements and taking the mean as the representative profile (see Warner et al., 2005; Talke et al., 2009; Schramkowski et al., 2015). This results in the following expression for the salinity profile

$$s(x) = 15\left[1 - \tanh\left(\frac{x - 55 \cdot 10^3}{26 \cdot 10^3}\right)\right]$$

The river discharge varies over the seasons, with an average summer discharge of 30 m³/s and an average winter discharge of 80 m³/s. Sediment concentrations found in the Scheldt are moderate, with near-surface concentrations only occasionally and locally exceeding 200 mg/l. Based on yearly-averaged data of the suspended matter concentration



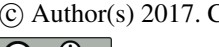

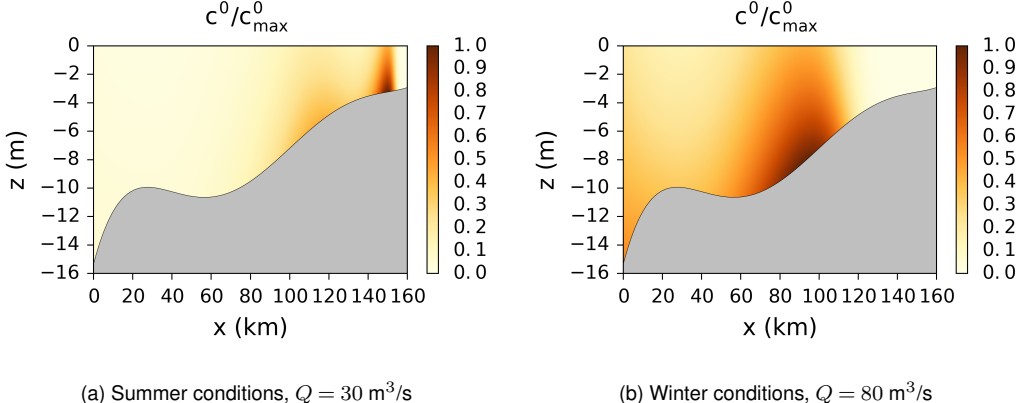

(a) Summer conditions, $Q = 30 \ \mathrm{m^3/s}$          (b) Winter conditions, $Q = 80 \ \mathrm{m^3/s}$

**Figure 11.** Tidally averaged sediment concentration for summer and winter conditions. Parameter values corresponding to the Scheldt Estuary

(Brouwer et al., 2016), the ETM is typically found between 100 and 140 km upstream from the mouth. However, monthly-averaged data indicate that for moderately high discharges, the ETM can be found around 60-70 upstream from the mouth or even disappear entirely. Under the assumptions described above, both the semi-analytical and the numerical solution methods may be used. Here we will use the semi-analytical method.

The model for the Scheldt Estuary is calibrated by tuning the partial slip roughness parameter $s_{f_0}$ to measured water level data. The calibration procedure is similar to that for the Yangtze River, but is not shown here (for details, see Dijkstra et al., 2016a). A best fit was found for $s_{f_0} = 0.0048$, which results in a good agreement with the $M_2$ water level and phase and $M_4$ phase, but leads to an overestimation of the $M_4$ water level.

Using the above parameter values and settings we compute the tidally averaged sediment concentration for average
summer and winter conditions, see Fig. 11. Since we are interested only in the location and relative magnitude of the ETM the concentration is scaled by the maximum concentration in the domain. It follows that, for average summer conditions, two ETM are present: a strong ETM near the weir at approximately 150 km and a weaker one at approximately 120 km. For average winter conditions the ETM is pushed in the downstream direction to approximately 100 km from the mouth. These results are in qualitative agreement with observations and thus suggest that the model captures the most important
physical mechanisms underlying ETM dynamics in the Scheldt Estuary.

In order to further investigate the underlying physical mechanisms of the ETM dynamics in the Scheldt Estuary, we look closer at the individual processes that contribute to the sediment transport. As explained in Section 4.5, iFlow allows investigating the transport contribution due to the individual contributions to the sediment concentration and the flow velocity. Five of the, in this case, most important transport contributions are shown in Fig. 12 for average summer
and winter conditions. The individual transport terms are scaled with the maximum value. Note that the total transport, obtained by adding all contributions, equals zero by definition of the morphodynamic equilibrium. For both summer and winter conditions the main up-estuary transport is due to the internally generated asymmetries of the velocity and depth





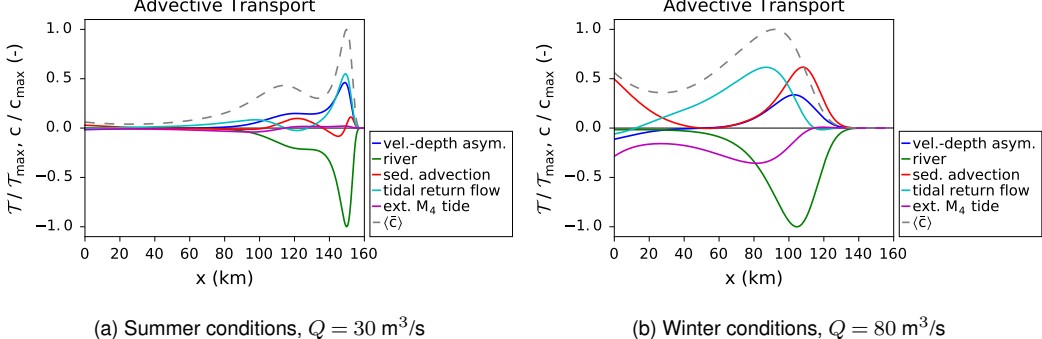

(a) Summer conditions, $Q = 30$ m$^3$/s    (b) Winter conditions, $Q = 80$ m$^3$/s

**Figure 12.** The five most important tidally averaged sediment transport contributions, rescaled with the maximum transport. Negative values indicates export, while positive values indicate import. Tot net transport of all terms added yields zero. The scaled tidally averaged depth-averaged sediment concentration (grey dashed line) is plotted in the background for reference. Parameter values corresponding to the Scheldt Estuary.

and tidal return flow tidal flow. During winter conditions, the spatial settling lag (i.e. sediment advection) is important as well. The down-estuary transport is mainly due to the river flow. The external $M_4$ tide additionally promotes export of sediment in winter conditions.

To illustrate iFlow's capacity to easily perform an extensive sensitivity analysis, we further analyse the influence of the external $M_4$ tidal component on the ETM dynamics in winter conditions. Even though it is not likely that this component changes on a short time-scale, it might alter under influence of climate change. Here, we select the $M_4$ phase for illustrative purposes only. The sensitivity study comprises 361 different values of the external $M_4$ tidal phase ranging between -180° and 180°. The results of all individual simulations are post-processed and the results are shown in Fig. 13. The result shows that the ETM can shift between approximately 70 and 110 km from the mouth depending on the phase of the external $M_4$ tide. The $M_4$ tidal transport induced maximum sediment export at a phase of approximately 50°, whereas it induces minimum export for a phase of approximately -100°. For phases between -60 and 140°, the model also indicates the existence of a concentration minimum. Such a minimum is characterised by a divergence of the sediment transport.

## 7 Conclusions

We have demonstrated that iFlow provides a flexible and versatile modular modelling environment for modelling flows and sediment transport in estuaries and tidal rivers. The model focusses on idealised approaches that allow the systematic analysis of physical processes and the sensitivity of these processes to model parameters. Due to the modular nature, iFlow offers a software environment where one can easily adjust the processes included in a simulation, thereby allowing to adjust the degree of complexity, computational time and ability to analyse the results to a specific application. The iFlow





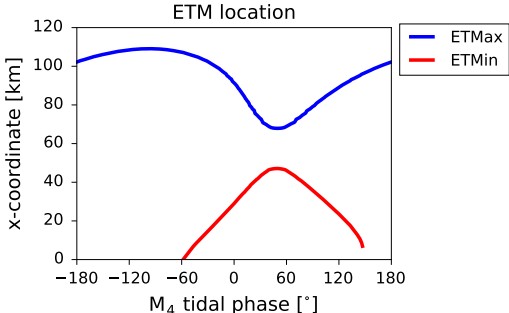

**Figure 13.** Sensitivity of the location of the ETM to varying external $M_4$ tidal phase in the Scheldt Estuary. The blue and red lines indicate a maximum and minimum in the sediment concentration in the model domain, respectively.

core supports these adjustments by automatically taking care of the communication between modules, order of modules and smooth coupling of modules that use different solution methods. iFlow version 2.4 additionally includes a number of standard modules especially designed to analyse individual processes affecting the flow and sediment transport

This has been illustrated in two applications of iFlow to the study of non-linear hydrodynamic interactions in the Yangtze
River and sediment trapping mechanisms in the Scheldt River. By a systematic approach of switching particular processes on and off and by the decomposition of the results according the forcing physical process the model allows for a unique insight into the physics of these systems. As iFlow allows for a quick set-up and calibration of a model and quick sensitivity study, the model is especially well suited to gain a first insight into the essential processes of a system and response of the system to changes. The comparison of the model results with observations in these systems should be mainly
interpreted qualitatively, focussing on the relative importance of processes, behaviour and sensitivity. Nevertheless, in the shown applications, there is a good quantitative correspondence between the model result and observations considering the degree of schematisation in the model.

Both shown applications used different modules and interactions, so that the model complexity suits the analysis relevant to the application. This extendibility, interchangeability and easy of analysis form the main ideas of iFlow. These
ideas are reflected in the architecture of the modular set-up, data management and within the modules offered within this version of the model. Wrapped around this is a set of tools and modules to support input, output and visualisation of the results to make the model user-friendly.

As the structure of iFlow can be adapted and modules can be added easily by new users, there is no such thing as a single iFlow model. Also the default provided modules for hydrodynamics, turbulence and sediment dynamics may
be replaced if this is useful for a particular application. For example, these modules may be replaced by a coupling to a complex model (e.g. as demonstrated for turbulence by Dijkstra et al., Manuscript submitted to JGR-Oceans) or observations. By coupling such module replacements to other modules one can construct unique model set-ups for studying a certain process or for comparing different model implementations within one modelling framework.





The future ambitions for the model involve further developments of modules for turbulence and morphology and for the transport of sediment, salinity and nutrients. Users are encouraged to contribute to this development by developing and sharing modules or sharing model applications.

## 8  Code availability

When using iFlow in any scientific publication, technical report or otherwise formal writing, authors are strongly requested to cite this paper and mention the name iFlow.

The iFlow code is property of the Flemish Dutch Scheldt Committee (VNSC) and is licensed under LGPL (GNU Lesser General Public License). In summary, this means that the code is open source and may be used freely for non-commercial and commercial purposes. Any alterations to the iFlow source code (core and modules) must be licensed under LGPL as
well. However, new modules or a coupling between iFlow and other software may be published under a different licence. Nevertheless users of iFlow are encouraged to make their own developed modules and model applications open source as well.

iFlow is written in Python 2.7 and the code is available through GitHub (https://github.com/YoeriDijkstra/iFlow). This repository contains the iFlow source code and the manuals.

*Acknowledgements.*  The developement of iFlow is funded by VNSC (http://www.vnsc.eu) through contracts 3109 6925 and 3110 6170 of the "Agenda for the Future" scientific research program that is aimed at a better understanding of the Scheldt Estuary for improved policy and management.





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



| Package | Module | Description |
|---|---|---|
| general | Output | save output variables for use within iFlow |
| | ReadSingle | load a single iFlow output file |
| | ReadMultiple | load multiple iFlow output files |
| | Sensitivity | Intelligently loop the simulation over any number of values of any number of variables |
| | ManualCalibrationPlot | Evaluate the result of a sensitivity analysis using a cost function that compares model results to data and plot the result |
| numerical2DV | RegularGrid | Create a 2DV standard grid and output grid. |
| | HydroLead | Leading-order hydrodynamics using fully numerical methods |
| | HydroFirst | First-order hydrodynamics using fully numerical methods |
| | HydroHigher | higher-order hydrodynamics up to any order using fully numerical methods |
| | HigherOrderIterator | Auxiliary module for higher-order computations (i.e. above first order) |
| | ReferenceLevel | Computation of a sub-tidal reference level based on the river-induced set-up |
| | SedDynamicLead | Leading-order sediment dynamics using fully numerical methods |
| | SedDynamicFirst | First-order sediment dynamics using fully numerical methods |
| | SedDynamicSecond | Second-order sediment dynamics restricted to river-induced resuspension of sediment, using fully numerical methods |
| | StaticAvailability | Sediment transport and trapping. Closure module for SedDynamicLead, SedDynamicFirst and SedDynamicSecond. |
| | SalinityLead | Dynamic leading-order salinity computation using fully numerical methods |
| | SalinityFirst | Dynamic first-order salinity computation using fully numerical methods |
| semi_analytical2DV | HydroLead | Leading-order hydrodynamics. Fully analytical in the vertical direction and numerical in the horizontal direction |
| | HydroFirst | First-order hydrodynamics.Fully analytical in the vertical direction and numerical in the horizontal direction |
| | SedDynamic | Leading-, first- and second-order sediment dynamics and transport/trapping using analytical solutions, but with numerical integration. The second-order sediment dynamics is restricted to river-induced resuspension. |
| analytical2DV | Geometry2DV | create a two-dimensional geometry with arbitrary depth and width |
| | SaltHyperbolicTangent | diagnostic (i.e. prescribed) well-mixed salinity field according to a $\tanh$ function |
| | SaltExponential | diagnostic (i.e. prescribed) well-mixed salinity field according to an exponential function |
| | TurbulenceUniform | Prescribed vertically uniform eddy viscosity and roughness |
| | TurbulenceParabolic | Prescribed eddy viscosity with a parabolic vertical profile and constant roughness |
| | KEFittedLead | Set of modules for a vertically uniform eddy viscosity depending on the local velocity |
| | KEFittedFirst | and depth, and for the roughness depending on the local velocity. The dependency |
| | KEFittedHigher | between the eddy viscosity and roughness is drawn from relations obtained from a |
| | KEFittedTruncated | $k - \varepsilon$ model. |

**Table 1.** List of modules included in iFlow version 2.4.





| Short name | Explanation | Order |
|---|---|---|
| **Hydrodynamics** | | |
| Tide | Tidal amplitude forced at the seaward boundary | 0 and 1 |
| River | Constant river discharge at the landward boundary | 0 (numerical) or 1 |
| Baroclinic | Forcing by the along-channel baroclinic pressure gradient | 1 |
| Advection | Effect of momentum advection $uu_x + wu_z$ | 1 |
| Tidal return flow | The return flow required to compensate for the mass flux induced by tidal correlations between the velocity and water level elevation | 1 |
| Eddy viscosity | Effect of higher-order eddy viscosity contributions. | 1 |
| Velocity-depth asymmetry | Correction for the alteration of the velocity profile due to the application of the no-stress boundary condition at $z = R$ instead of the real surface $z = R + \zeta$ | 1 |
| **Sediment dynamics** | | |
| Erosion | Local resuspension at the bed | 0 and 1 |
| Spatial settling lag | Effect of sediment advection $uc_x + wc_z$ | 1 |
| Surface correction | Correction because the transport across the time-dependent water surface is specified at $z = R$ instead of the real surface $z = R + \zeta$ | 1 |
| Fall velocity correction | Effect of higher-order variations of the fall velocity | 1 |
| Mixing correction | Effect of higher-order variations of the eddy diffusivity | 1 |

**Table 2.** Separate forcing mechanisms to the water and sediment motion and the order at which these mechanisms appear.

| | semi-analytical | numerical |
|---|---|---|
| Orders hydrodynamics | Leading and first | Any |
| Orders sediment dynamics | Leading and first | Leading and first |
| Eddy viscosity/diffusivity | Vertically uniform, sub-tidal in leading order and $M_2$ frequency in first order | Vertical variations and leading-order and first-order time variations allowed |
| Bottom boundary condition | Partial slip with constant roughness | Partial slip with time-varying roughness or no-slip |
| Leading-order forcing tidal components | $M_2$ | any |
| First-order forcing tidal components | $M_4$ | any |
| River discharge | first order | leading or first order |
| Fall velocity | Vertically uniform, sub-tidal in leading order and none in first order | Vertical variations and leading-order and first-order time variations allowed |

**Table 3.** Allowed forcing and turbulence options in the semi-analytical and numerical solution methods.