# Peer review of "The iFlow Modelling Framework v2.4. A modular idealised process-based model for flow and transport in estuaries."

_Geoscientific Model Development, 2017_

## Referee Comment (RC1) · Anonymous Referee #1 · 16 Mar 2017

This paper introduces a framework iFlow to systematically study the width-averaged water motion and sediment transport. The framework is built in a modular structure combining the strengths of idealised analytical models and complex numerical process-based models. This structure is easily extendable to allow for more processes to be included. It offers an option for both analytical and numerical methods for different systems even those with complex non-linear processes. Also, different turbulence closure schemes are made available within the framework. This framework is evaluated using two examples (the Yangtze and Scheldt estuaries), showing its capability of investigating the tide-river interactions and sediment transport contributions due to different physical processes, and its sensitivity to tidal phases. They also found that inclusion

of reference levels in calculating estuarine water motion helps to get more realistic solutions.

The framework introduced in this paper can be used to better understand the dominant physical processes of estuarine hydrodynamics and sediment transport. The effort of building up this framework and drafting the detailed manuals is highly appreciated, which will greatly help future users to run the model and promote the study of the dominant physics of estuarine dynamics. However, the paper seems to be not very complete in terms of demonstrating the solution methods and how different processes are separated within those methods. This could lead to difficulties for readers to see the strength of the model and limit potential users to understand their results and extend the framework (e.g., include more processes).

Major concerns:

1, Since being able to identify contributions of different physical processes is one of the main strengths of iFlow, the solution method should be more clearly demonstrated in the paper especially in terms of how different processes are separated. These details seem to be included in the supplemented manual, but including/summarizing the important details in the paper will make it more complete and clear to readers without reading extra tens of pages in the manuals. For example, processes listed in Table 2 could be explained in more detail and some literature using the same approach could be cited here. Section 4.4 seems to suffer the same problem, please add explicit expressions or at least cite existing papers where those detailed expressions are described, for example, in Chernetsky et al (2010).

2, The paper seems to include very heavy technical details about modular structures and running management (in section 2.1-2.2). To make the paper more concise, I think those details should be greatly reduced and preferably integrated in the solution method section (or move to the manual).

3, The perturbation method used within this framework has been extensively used in

previous studies (Chernetsky et al 2010, McCarthy 1993, Wei et al 2016). However, the main assumptions used in this framework are not very clearly outlined in the paper. What main assumptions are used in the model? What are the (unresolved) potentially important processes to sediment transport which need to be added in the framework in the near future? For instance, since iFlow is width-averaged, longitudinal-vertical processes are focused while lateral processes are assumed to be insignificant and not resolved. Also, river branching, which is important in the Yangtze estuary (the 1st study site in the paper), is not included.

4, Different turbulence closure schemes are available in the framework, which one would you recommend to future users for different purposes? A short overview of the strength and weakness of different schemes would make these options more rational and straightforward. On page 27, the authors write "Here we will use the semi-analytical method". Again, why not the numerical method here?

5, This paper has a few sections lumping different (long) contents, which makes it difficult to locate interesting/important information. For example, in section 4.5, semi-analytical method and numerical method could be separated into different subsections; in section 6.1 (and 6.2), the model settings, and main findings could be put in different subsections.

minor comments:

1, Section 6.1 mainly focuses on the river effects on the tide, while the tidal effect on river is not investigated. So I think the section title "tide-river interaction" should be modified.

2, Page 28, the authors wrote the influence of "climate change", do you mean sea level rise? Try to be more specific.

3, Table 3, does it work only for M2, M4? Can it be M1, M2, for example?

4, Page 2, by increasingly => by including

5, Page 18, estimated numerically by from => estimated numerically from

6, Page 26, Eq (20)-(21) => Eqs (20)-(21)

7, Page 27, 60-70 => 60-70 km

8, Page 27, the unit of $s_{f_0}$ should be added

9, Page 27, two ETM's

---

## Referee Comment (RC2) · HHG Savenije (Referee) · 18 Apr 2017

To me this looks like very useful software to analyse tidal motion and sediment dynamics by a 1DV model. The charm of the model is that it allows to derive analytical solutions of different orders of approximation, by which the analyst can see what the contribution of higher order effects is. The paper is well-prepared and the software is described clearly. The fact that it is open-source and modular makes it into an interesting basis for (other) researchers to add further applications.

For instance the way in which the salinity is calculated is rather superficial: merely by fitting a hyperbolic function. It seems to me that this can be done a lot better, since the software already determines a lot of hydraulic parameters that are needed to use

more physics-based analytical equations for the salinity distribution. It would be useful to explore more physics-based equations for salinity dispersion in real estuaries. But I am sure it is the intention of the developers to facilitate such an expansion.

Minor comment:

p.22, line 2: This equation for the width looks very strange to me. It seems to me that two of the three numbers in the denominator are irrelevant and that the value of 3.2 is appropriate. I am sure that the authors are aware of the meaning of significant decimals. Further, it is easier to divide the two terms in the numerator directly by 3.2 and make the equation simpler.

---

## Author Comment (AC1) · 15 May 2017

Dear editor, dear reviewers,

Thank you for the reviews. We will address the comments of the reviewers individually below

[Figure]

**Anonymous reviewer 1**

**Major concern 1:** We agree that it was not clearly mentioned how the perturbation method allows for the separation of processes and agree that at least the main ideas behind this should be illustrated in the paper. Therefore we have extended Section 4.3 (perturbation method) with more equations and explanations for the leading- and first-order hydrodynamics, with explicit connections between the method of decompostion (linearity and principle of superposition), forcing terms in the equations and the terms in Table 2. For the sediment dynamics the text has been expanded to show the connection between the forcing terms in the equations and Table 2, by explicitly referring to the forcing term. The principle is the same for the sediment equations as for the hydrodynamics. For brevity we still refer to the manual for the ordered sediment equations.

Section 4.4 (harmonic analysis) has been restructured with a bit more elaboration. However, we have opted not to include full equations here, since we feel this is not necessary for understanding the main text and would lead to repetitiveness and too much mathematics. References to Chernetsky (2010) and the manuals have been added.

**Major concern 2:** To our view the iFlow model makes two major new contributions: 1) the modular approach allows for much more flexibility and extendability of the model and is much more user-friendly and 2) on content it contains a vast set of extensions of the model of Chernetsky (2010). The first is to our view important for the future use and development of the model and in the decision of other researchers to use or develop it. The modular structure, running procedure and data management are therefore essential parts of the iFlow model. Since this paper and this journal are aimed at presenting model/software development, we have deliberately aimed at a comprehensive discussion of the modular structure and running management in Sections 2.1 and 2.2. For us, this, together with the mathematical analysis method, is the core of the paper and

of paramount importance for providing readers with an idea of what this software looks like and what it can do. Therefore we have chosen to keep these sections.

**Major concern 3:** In our view, the main assumptions behind the perturbation model were already listed explicitly in Section 4.2. The assumptions on the forcing terms in the semi-analytical version of the model were omitted in favour of generality and may therefore have been unclear. We have changed this and now explicitly list the assumptions on the forcing for the semi-analytical model in Section 4.5. The consequences to the model analysis when choosing this forcing have also been more clearly stated in Section 4.5

Concerning the unresolved processes: being an idealised model, iFlow will always omit several processes. The relevant question is indeed whether the most essential processes for the investigated phenomena are taken into account. This differs from one estuary to the other and from one investigated phenomenon to another. It is therefore impossible to say, in general, what processes should be added to the model in the future. Concerning the specific cases presented in the paper, we show that there is a good comparison between observations and model results. Although this cannot count as a full proof that iFlow includes all essential processes in these systems, it at least provides confidence that the most important processes are accounted for.

**Major concern 4:** Recommendations on which turbulence model to use have been added to section 5.1. In the case studies we choose the semi-analytical method whenever it is possible to apply it, since it is faster and more accurate than the numerical method. In the Yangtze case, the numerical method is used because the semi-analytical method cannot be used . This is now mentioned explicitly in the introduction to Section 6.

**Major concern 5:** Section 4.5 has been separated into a section on analysis of sediment transport and a section on the semi-analytical and numerical methods. As the long discussion on sediment transport is now separate, the part on the semi-analytical

method is short and it makes sense to us to group it with the explanation of the numerical method.

Section 6.1 and 6.2 have been splitted as suggested.

**Minor comments**: 1. the title 'tide-river interaction' has been adjusted to 'river-induced modification of the tidal propagation'

2. reference to climate change has been removed

3. The method can be applied to $M_1$, $M_2$ etc. as well. This is now mentioned in section 4.4. By default we will assume that the base frequency is that of the $M_2$ tide, as this is the most common case.

4.-7. typo/spelling mistake resolved

8. unit has been added

9. ETM has been changed to ETMs.

**Reviewer 2: Dr. H.H.G. Savenije**

As noted by the reviewer, the iFlow model is indeed developed in such a way that it can be extended easily to include other salinity formulations. Right now, the model offers simple hyperbolic and exponential profiles, as well as a somewhat more involved physics-based model of Wei et al (2016), see Section 5.2 of the manuscript. These salinity models have been sufficient for our applications to date, but might need to be extended in the future. Indeed the model already determines many hydrodynamic parameters, so that there is an opportunity for us or other developers to do this when a possibility and relevant case arises.

Concerning the minor comment: the equation on p22 was missing some $x$-

dependencies. This has been corrected.

Yours sincerely,

Yoeri Dijkstra,
also on behalf of my co-authors

————————————————